

# Unmanned Aerial System nadir reflectance and MODIS Nadir BRDF-Adjusted surface Reflectances intercompared over Greenland

Burkhart, John Faulkner[1,2], Kylling, Arve[3], Schaaf, Crystal B.[4], Wang, Zhuosen [5,6], Bogren, Wiley[7], Storvold, Rune[8], Solbø, Stian[8], Pedersen, Christina A.[9], and Gerland, Sebastian[9]

[1]Department of Geosciences, University of Oslo, Oslo, Norway
[2]University of California, Merced, CA, USA
[3]Norwegian Institute for Air Research, Kjeller, Norway
[4]School for the Environment, University of Massachusetts Boston, Boston, MA, USA
[5]NASA Goddard Space Flight Center, Greenbelt, MD, USA
[6]Earth System Science Interdisciplinary Center, University of Maryland, College Park, MD, USA
[7]U.S. Geological Survey, Flagstaff, AZ, USA
[8]Norut-Northern Research Institute, Tromsø, Norway
[9]Norwegian Polar Institute, Fram Centre, Tromsø, Norway

*Correspondence to:* John F. Burkhart (john.burkhart@geo.uio.no)

**Abstract.** Albedo is a fundamental parameter in earth sciences. Many datasets are developed from the MODIS BRDF/Albedo (MCD43) Algorithms. While derivative albedo products have been evaluated over Greenland, we present a novel direct intercomparison with nadir surface reflectance collected from an Unmanned Aerial System (UAS). The UAS was flown from Summit, Greenland on 200+ km transects coincident with the MODIS sensor overpass on board the Aqua and Terra satellites on

5  5 and 6 August, 2010. Clear sky acquistions were available from the overpasses within two hours of the UAS flights. The UAS was equipped with updward and downward looking spectrometers (300-920 nm) with a spectral resolution of 10 nm allowing to integrate directly to the MODIS bands 1, 3 and 4. The data provides a unique opportunity to directly compare UAS nadir reflectance with the MODIS Nadir BRDF-Adjusted surface Reflectance (NBAR) products. The data show UAS measurements are slightly higher than the MODIS NBARs for all bands, but agree within their stated uncertainties. Differences in variability

10  are observed as expected due to different footprints of the platforms. The UAS data demonstrate potentially large sub-pixel variability of MODIS reflectance products and the potential to explore this variability using the UAS as a platform. It is also found that even at the low elevations flown typically by UAS, reflectance measurements may be influenced by haze if present at and/or below the flight altitude of the UAS. This impact could explain some differences between data from the two platforms.

## 1  Introduction

15  Albedo, the ratio of reflected to incident energy at the surface of the earth, is a fundamental parameter in energy balance computations and therefore any prediction of climate must account for albedo through a parameterization process (Henderson-Sellers and Wilson, 1983). Generally climate models rely on simplified estimations of albedo as single value climatological





means of broadband albedo that are a function of seasonal changes in surface characteristics and the presence of snow (Curry and Schramm, 2001). For modeling snow and ice melt processes on the earth's surface, albedo is a critical parameter, providing the most coarse adjustment with respect to available energy to drive melt. Satellite instruments play a critical role providing a characterization of albedo of the surface of earth that is relevant for climate and earth system modeling.

Several studies have evaluated presently available satellite products and compared these products with ground-based observations. Stroeve et al. (2005) and more recently Stroeve et al. (2013) have carefully evaluated the MODIS albedo products over Greenland. This recent body of work has been largely spurred by the recent melt events on the Greenland Ice Sheet. These events, recorded by MODIS satellite observations, have been linked to albedo feedback stemming from thermodynamic processes (Box et al., 2012). Dumont et al. (2014), Goelles et al. (2015), and Keegan et al. (2014) attribute some of the darkening
to deposition of soot from forest fires, pollution and dust, while others have linked the changes predominately to delivery of warm water vapor and low-level clouds (Bennartz et al., 2013; Miller et al., 2015).

Due to the impact on snow and ice albedo, the International Panel on Climate Change (IPCC) identified black carbon on snow as an important process driving changes in the cryospheric energy balance with significant associated uncertainty (Stocker et al., 2013). These findings are based on research that has focused on the theoretical response of snow to black carbon deposition
(Warren and Wiscombe, 1980; Hansen and Nazarenko, 2004). Subsequent studies have shown that small changes in snow albedo globally, may have significant impact on the top of atmosphere forcing and could be driving a component of the Arctic warming witnessed today (Flanner et al., 2009). However, presenting a distinct challenge, Warren (2012) suggests that the changes anticipated from this effect are below the present-day measurement capabilities.

Ground based measurements with satellite borne sensors are critical to assess accuracy of the observations, and particularly
to understand the variability that may be missed by different sensor footprint scales. A seminal study is that of Salomonson and Marlatt (1971) who conducted an evaluation of surface reflectance conditions for application to retrievals from the Medium Resolution Infrared Radiometer (MRIR) instrument aboard the Nimbus II and III satellites. The study focused on the measurement of bi-directional reflectance distribution function (BRDF) over a variety of terrestrial surfaces. Appreciable anisotropy in all the surfaces was found, leading to the conclusion of the importance of using a BRDF model for satellite retrievals – a
standard application today (Schaaf et al., 2002; Jin, 2003; Román et al., 2009; Ju et al., 2010; Wang et al., 2014).

Wright et al. (2014) intercompare ground based spectral observations from an ASD Field Spectrometer at Summit station with both the MODIS C5 and C6 data, and show a marked improvement of the MODIS C6 retrieval. Prior studies relied predominately on existing GrIS fixed station data from the GC-NET network of Automatic Weather Stations (Box et al., 2012; Stroeve et al., 2005, 2006, 2013). Other prior investigations have also used ground based observations, but to our knowledge
most recent investigations have compared ground based measurements of albedo with satellite albedo products. While this is valuable, one must recognize that albedo products are developed through a processing chain of models and therefore do represent a direct measurement, making intercomparisons complicated.

Recently, the advent of relatively low-cost Unmanned Airborne Systems (UAS) has driven created a rush to utilize this novel platform to provide unique datasets otherwise unobtainable without manned flight. Furthermore, UAS provides a unique niche
in the ability to characterize cryospheric surfaces in relatively localized regions at higher resolutions than may be possible with



traditional aircraft and certainly offers the potential to extend the observational range of a traditional ground-based campaign Bhardwaj et al. (2016). In one of the first applications of UAS for cryospheric characterization, Hakala et al. (2014) demonstrated the potential for BRDF measurements from a simple quad-copter. Immerzeel et al. (2014) used UAS to characterize glacial dynamics in the Himalaya, while numerous other have recently applied structure from motion photogrammetry to sev-

eral applications related to snow and ice surfaces Jagt et al. (2015); Ryan et al. (2015); Rippin et al. (2015). Most recently, Ryan et al. (2016) has attempted to directly measure albedo from a UAS using pyronometers on board a fixed wing platform on the perimeter of Greenland.

Herein we provide a first-of-a-kind, "apples to apples" evaluation of the accuracy of the MODIS Nadir BRDF-Adjusted Reflectance (NBAR) retrievals through intercomparison with reflectance observed from an UAS platform near the summit of

the Greenland Ice Sheet. The advent of UAS presents an immense opportunity to spatially assess the accuracy of satellite sensors, versus simple validation against ground point observations, but as discussed in this work, has a host of complications as well. Our analysis provides data for two transects on separate days covering over 200 km of ground designed to coincide with the near nadir subtrack of the MODIS instrument overpasses.

In this study, spectral reflectance measurements made from a UAS flying in the dry snow region near the summit of the

Greenland Ice Sheet are used to evaluate sub-pixel scale variability of the MODIS NBAR retrievals. The campaign was conducted in 2010. Due to the pristine nature of the snow pack in this area, and the limited influence of aerosols and warm temperatures, albedo and reflectance variability in this region is expected to be less than 10%, and potentially as low as 3%, within the 5% stated accuracy of the MODIS datasets. The standard MODIS products retrieve narrowband reflectance and then use a narrowband-to-broadband algorithm to convert the discrete narrowband measurements into a broadband albedo (Schaaf

et al., 2002; Stroeve et al., 2005). For direct comparison with MODIS, the UAS observations here are converted into shortwave narrowband reflectance values to coincide with MODIS bands 1, 3, and 4 (see Sect. 5).

The research presented herein was conducted as part of the Norwegian "Variability of Albedo Using Unmanned Aerial Vehicles" (VAUUAV) project. A guiding objective of the research was to evaluate whether present day satellite observations allow the capacity to evaluate albedo variability across a cryospheric landscape and provide input for validation of theoretical

modeling. However, in our work, we discovered the application of UAS to obtain a relevant measure for validation of MODIS datasets is greatly complicated by aspects of the platform that to date have not been addressed, particularly with respect to albedo. Therefore, as we have the capability, we have chosen to conduct the intercomparison with reflectance – providing a more direct evaluation of the platform capabilities. Further we have attempted to address several of the complex issues that result from the UAS in this analysis and otherwise highlight the potential for uncertainty in the observations. In Sect. 2 we

present the UAS measurement platform. Section 3 describes radiative transfer calculations that were used in the study, while Sect. 4 describes some aspect of the data selection. In Sect. 5 we describe the MODIS data used for this study. A comparison and discusson of the MODIS and UAS data are presented in the Sect. 6. We summarize and conclude our main points in the conclusion.



## 2 Surface reflectance measurements from an Unmanned Aerial System (UAS)

The Cryowing (see Sect. 2.1) UAS performed several flights during the summer of 2010 in the region of Summit, Greenland
at the Greenland Environmental Observatory, Summit (http://www.geosummit.org). The flights were designed to measure the
downwelling irradiance and the upwelling nadir radiance as discussed further in Sect. 2.2.

Two of the flights were specifically designed to be closely aligned with MODIS overpasses and were flown as close in time
as possible to the satellite overpasses. On 5 and 6 August 2010, the UAS completed a flight pattern with coverage over a region
which was nearly coincident with the MODIS sensor overpass on board the Aqua and Terra satellites. On both days, clear
sky acquistions are available from overpasses within two hours of the UAS flights. The flight pattern covered 210 km ground
distance and was completed autonomously for a duration of over two hours. From the UAS observations we develop a nadir
dataset suitable for direct comparison with the MODIS NBAR products with a reduced reliance on a complex model chain.

### 2.1 The Cryowing UAS

The Cryowing UAS is an autonomous fixed-wing airborne sensor platform developed in Norway. It has a maximum takeoff
weight of 30 kg, payload capacity of 15 kg including fuel, and a wingspan of 3.8 m. The Cryowing is powered by a two stroke
engine, fueled by a petrol-oil mixture. The normal cruising speed is 100–120 km h$^{-1}$, with a range of up to 500 km or 5 hours
flight. The Cryowing has a 2500 m dynamic altitude range, with a 5000 m absolute altitude cap. A dedicated GPS independent
from the payload is used for navigation and autopilot control. While the Cryowing is capable of autonomous control for the full
period of a flight, in practice a skilled technician is present to control launch and landing via radio control. Once stable flight
is achieved, communication with the UAS is maintained for the duration of the flight using radio modem or Iridium satellite
modem.

A standard suite of instruments is deployed on the Cryowing. This includes a meteorological package, two inertial measure-
ment units (IMU) and two GPS systems, as well as the computer and communications systems responsible for flight control
and ground station contact. The meteorological package measures air pressure, temperature, and humidity; while the flight
computer and systems record aircraft position and altitude. Position is recorded as altitude in meters along with latitude and
longitude, while attitude is recorded in quaternion form. In this analysis platform attitude was converted to the azimuth and
zenith angles for subsequent radiative transfer modeling as discussed in Sect. 3. Position and attitude variables are recorded at
a frequency of 100Hz by the IMU.

### 2.2 UAS-based surface reflectance

The *in situ* observations presented in this study provide the downwelling irradiance ($E(\theta_i)$) and the upwelling radiance
$L_r(\theta_i, \phi_i; \theta_r, \phi_r)$ with a field-of-view of 7° in the direction $\theta_r = \pi/2, \phi_r = 0$ for incident zenith (azimuth) angle $\theta_i$ ($\phi_i$).
The nadir reflectance measured by the UAS is

$$\rho = \frac{\pi L_r(\theta_i, \phi_i; \pi/2, 0)}{E(\theta_i)} \tag{1}$$



which may be directly compared with the Nadir BRDF-Adjusted surface Reflectances (NBAR) from MODIS (Schaaf et al., 2002). All wavelength dependence in Eq. 1 has been omitted for clarity.

Our instrument measures spectral reflectance from 320 nm to 950 nm with 3.3 nm per pixel resolution with a 3 nm over-sampling making an effective 10 nm resolution, allowing us to integrate across the MODIS bands 1, 3 and 4 for a direct intercomparison. The spectral response for MODIS bands 1, 3, and 4 are shown in Fig. 1 and the wavelengths covered by these bands are given in Table 1.

In the VAUUAV payload configuration, the Cryowing is equipped with two Trios Ramses spectroradiometers to measure reflected and incoming radiation in the visible spectrum. The upward facing sensor, measuring incoming radiation, has a cosine corrected foreoptic made of synthetic fused sylica, transparent to 190 nm, to measure full sky hemisphere irradiance. The nadir facing sensor has a Gershun Tube restricted 7° Field Of View (FOV) foreoptic, measuring reflected radiance emanating from a footprint beneath the plane. At a cruise altitude of 250 m the footprint is on the order of 30 m in diameter.

Details of the TriOS sensors and the configuration used can be found in Nicolaus et al. (2010). For completeness, we describe the essential characteristics of the spectral radiometers here. The TriOS RAMSES ACC-2 VIS hyper-spectral radiometers are based on a miniature spectrometer with a wavelength range from 310 to 1100 nm, and spectral resolution and accuracy of 3.3 and 0.3 nm, respectively. TriOS uses the VIS/NIR specification of the spectroradiometers (wavelength range from 360 to 900 nm) to post-calibrate the instruments to a wavelength range from 320 to 950 nm. Software controlling the instruments enables an automatic adjustment of the integration time for each measurement, ranging between 4 and 8192 ms.

There are two versions of the sensors, one containing an inclination and pressure sensor, and one without. For our purposes, these additional components were not required, as a part of the standard suite of measurements aboard the UAS includes highly accurate inclination from the IMU. As the sensors were initially designed for water quality applications, they are built to be water resistent to a depth of 300 m. This creates additional weight due to the robust design of the casing and sealed body of the sensors. In order to use the sensors in the UAS, modifications from the sensors described in Nicolaus et al. (2010) were required. To reduce the length and weight of the sensors as available from the manufacturer (and described in (Nicolaus et al., 2010)), the solid steel casing was removed and replaced with a light weight aluminum version. This reduced the weight of the sensors significantly from the initial 833 g, to less than 400 g.

For comparison with the MODIS NBAR data (see Sect. 5), the UAS spectra were multiplied with the MODIS spectral band functions (Fig. 1) and the respective NBARs calculated according to Eq. 1.

## 2.3 Radiance offset correction

To establish the relative sensitivity of the radiance and irradiance sensors, measurements were made on the ground with the two UAS sensors co-located together with a reference irradiance sensor looking skyward. The relative sensitivity of the radiance and irradiance sensors was calculated and a third order polynomial was fit to this ratio in the wavelength region relevant for comparison of UAS and MODIS data. All measured radiance spectra were corrected for the wavelength dependent offset using the polynomial fit.





## 2.4 Cosine error correction

The uplooking sensor measures the irradiance, requiring a detector with a hemispheric cosine response foreoptic. In reality the angular response of cosine detector deviates from a cosine shape. Cosine error corrections have been thouroughly investigated for UV spectrometers (see for example Bais et al. (1998)). The cosine error correction depends on the atmospheric state when
the measurements were made. The deviations typically become larger as the incidence angle increases implying that measured irradiance is underestimated compared with an instrument with a perfect angular response. This underestimate may be corrected for providing that the sky conditions during the measurements are known and that the angular response of the instrument is known (Bais et al., 1998).

    The TriOS sensors are laboratory certified and have undergone calibration by the manufacturer prior to each field season.
For the zenith angles encountered during the flights ($<70°$) we expect the deviation from a perfect cosine response to be less than 2%.

## 2.5 Angular sensitivity

The uplooking detector must be properly levelled to allow accurate measurements of the downwelling irradiance (Bogren et al., 2016). This may be achieved by for example stabilizing the measurement platform Wendisch et al. (2001) or by mounting the
instrument with a tilt such that the instrument is levelled during flight. The latter approach was adopted with the UAS, however, this requires that the platform is stable during flight. To estimate the effect of angular changes on a fixed detector, radiative transfer simulations were performed as described below in section 3. The roll angle of the aircraft, and thus the detector, was changed between 0 and 10° while the yaw angle was changed from from 0 to 360°. The radiation field was simulated for a cloudless sky over a snow covered surface. The response relative to a levelled detector is shown in Fig. 3 for a solar zenith
angle of 55.66° and azimuth 0°. If the detector has a roll angle of 10.0° and yaw angle 90° with respect to the sun implying that it is facing away from the sun, the detector will measure only about 80% of the radiation of a leveled detector.

    Similarily, a detector shifted such that it faces the sun will overestimate the radiation compared to a leveled detector. The results presented in Fig. 3 are for a cloudless sky. For an overcast sky, with the aircraft flying below the cloud, the change in angular response is negligible with given azimuth and roll angles Bogren et al. (2016). The effect of roll and yaw angles on the
measurements will change with solar zenith angle, surface albedo, sky conditions and wavelength. As such they are challenging to correct for when the aircraft is moving around due to changes in the flying directions or changing wind conditions. For the analysis below, UAS data was screened and selected for stable flight conditions. In addition a tilt correction was applied to the direct portion of the irradiance impinging the upward facing sensor. As presented by Bogren et al. (2016) the response of a sensor tilted $\theta_t$ degrees and rotated $\phi$ degrees relative to the sun is

$$R^t(\theta_t, \phi) = \cos(\theta_0 - \theta_t \cos(\phi))$$ (2)





where $\theta_0$ is the solar zenith angle. For a levelled sensor $R^l = \cos(\theta_0)$. The tilt error correction is largest for the direct part of the irradiance and negligible for the diffuse part (Bogren et al., 2016). We thus tilt correct the measured downwelling irradiance $E_m$ as follows

$$E = fE_m R^l / R^t + (1-f)E_m. \tag{3}$$

Here the first term on the left side is the tilt correct direct contribution and the second term is the uncorrected diffuse contribution. Furthermore, $f$ is the wavelength dependent direct/global irradiance ratio. It was estimated by the libRadtran model, described in section 3, to be 0.98, 0.92 and 0.85 for MODIS bands 1, 4 and 3, respectively.

Due to dismounting and remounting for maintenance, or from the thrust of the catapult at launch, the instrument package may become slightly disoriented. Thus $\theta_t$ and $\phi$ may be offset. During analysis of the data best results were obtained by reducing $\theta_t$ by 0.7° and adding a 10° azimuth offset.

## 2.6 Atmospheric corrections

During the flights the altitude of the UAS varied between 270 and 320 meters above the surface. The atmosphere between the surface and the aircraft may influence the aircraft nadir measurements. To estimate the impact of the intervening atmosphere UAS radiance and irradiance spectra were simulated for noon (solar zenith angle of 55.66°) at Summit for elevations between 270 and 320 m.a.g.l. in steps of 10 meters. The simulated spectra were multiplied with the MODIS band 1, 3 and 4 response functions (Fig. 1) and the corresponding UAS nadir measurements were integrated to the corresponding narrow bandwidths. It was found that the atmosphere between the aircraft and surface caused less then 0.2% changes in the band 1 nadir reflectance and less then 0.04% difference in the band 3 and 4 nadir reflectance. Thus, the nadir reflectance derived from the UAS were not corrected for the intervening atmosphere.

## 2.7 Error estimate

Estimates of the measurement error is inherently difficult to make. Both because they are difficult to do and because they require time resources often not available in the field. However, best estimates of the measurment error due to various sources are assumed and used to calculate a total error as summarized below.

Ideally NBAR measurements should be made over flat surfaces. As shown in (Siegfried et al., 2011) the region around Summit is sufficiently flat with a slope of less than 2 m km$^{-1}$ in the east-west direction and less than .5 m km$^{-1}$ in the north-south orientation. Our flights were further north from Summit than measured by Siegfried et al. (2011), but data from available digital elevations maps from the Greenland Ice Mapping Project Howat et al. (2014) confirm the area covered by the UAS flights is indeed flat. However, small scale wind-blown snow feature can not be ruled out. This is potentially the largest source of error in the analysis. Sustruggi structures on the snow surface can cause strong scattering and geometric optical effects. This variability is difficult and complex to resolve and adequately model. However, effects of scattering will be



reduced by integrating over the footprint of the measurement. An uncertainty of 0.5% is assumed due to measurements over non-flat surfaces.

The offset between the up- and down-looking sensors was measured and corrected for as explained in section 2.3. A remaining error of 0.2% is assumed for the offset correction.

The tilt error has been corrected for as described in section 2.5. The attitude is specified to have an uncertainty of 2%. The uncertainty in the data due to remaining tilt error and assumption about the direct/global radiation ratio is thus taken to be 2%.

According to the manufacturer the cosine error is better than 6-10%, depending on wavelength while for the 7° detector the angular response is better than 6%. The cosine error typically increases with zenith angle in addition to wavelength. The error will thus be largest for large solar zenith angles as found at high latitude. No cosine response measurements were available for a detailed assessment of the cosine error. A 2% cosine error correction has been applied to the uplooking sensor, section 2.4. A 2% cosine error uncertainty is assigned to the measurements. The downlooking sensor is exposed to diffuse radiation and the error in the angular response is of less concern.

For an integration time of 8 s the manufacturer gives a noise equivalent irradiance (NEI) of 0.4 $\mu$Wm$^{-2}$nm$^{-1}$ at 400 and 500 nm, and 0.6 $\mu$Wm$^{-2}$nm$^{-1}$ at 700 nm for the cosine response detector. For the 7° detector the NEI is 0.25 $\mu$Wm$^{-2}$nm$^{-1}$. The integration during the flights were shorter, thus a conservative estimate of the NEI during the flights is 0.5%.

We assume that all errors are independent of wavelength. Squaring the errors give a total error in the UAS reflectance of 2.9%.

## 3   Radiative transfer simulations

As a part of the data reduction and analysis process, we conducted radiative transfer simulations. These were performed to test the UAS sensitivity to changes in pitch, roll and yaw angles (section 2.5 above), and to simulate cloudless shortwave broadband radiation at Summit (section 4 below). The libRadtran radiative transfer package was utilized for these calculations (Mayer and Kylling, 2005; Emde et al., 2016). The molecular absorption was parameterised with the LOWTRAN band model (Pierluissi and Peng, 1985), as adopted from the SBDART code (Ricchiazzi et al., 1998). The C version of the DISORT radiative transfer solver (Stamnes et al., 1988; Buras et al., 2011) was utillized. The snow albedo model Wiscombe and Warren (1980) as implemented in the libRadtran software package, was used to calculate the spectral surface reflectance as shown in Figure 9. The sub-arctic summer atmosphere (Anderson et al., 1986) was used and the surface altitude set to 3126 m.

## 4   Measurements selected for analysis

On 5 and 6 August, 2010, the UAS flew a 210 km pattern designed such that one of the flight legs would be centered on the MODIS granule as close in time as possible to the MODIS overpass. The flight pattern flown is shown in Fig. 2. On Aug 5 the sky at Summit was overcast with some blue patches at take-off. The cloud deck thickened during the flight to a uniform diffuse cover. This development is readily visible in the global shortwave measurements recorded at Summit (blue line, Fig. 4).



The measured shortwave radiation is clearly below the cloudless simulated shortwave radiation and its behavior indicates the presence of clouds. The wind was blowing from the south-east with speeds around 6-7 m/s (Table 2). On the $6^{th}$ the sky was mostly clear during most of the flight with some thin layer cirrus forming at the end of the flight. The wind was more gentle on the $6^{th}$ with speeds increasing from about 1.5 to 3.5 m/s during the flight. At the same time the wind direction changed from south-east to a more southerly direction.

The pitch and roll angles of the UAS had non zero offsets indicating that the aircraft was flying in a non-leveled manner due to impact of wind, fuel-load and placement of the center of mass. The offsets varied between the various flights and are given in Table 2. UAS data that are within $\pm 0.5°$ of the mean pitch and roll angles were included in the analysis. Furthermore, due to the sensitivity of the measurements to the orientation of the uplooking instrument discussed above, only data for which the yaw angle was stable were included in the analysis. For 5 and 6 Aug. the instrument azimuth (blue dots) and the corresponding tilt correction (red dots) are shown in the top panel of Figs. 5 and 6, respectively.

MODIS band 1, 3 and 4 NBARs (see Sect. 5) were extracted from the MODIS pixels that coincide with the Cryowing UAS data points. The blue, green and red MODIS NBARs are shown in the second, third and fourth panels respectively, of Figs. 5 and 6. Data that are flagged as high quality are in black while lower quality flagged MODIS data are in yellow.

## 5 MODIS Nadir BRDF-Adjusted Surface Reflectance (NBAR) measurements

The MODIS NBAR product from MCD43 Collection 6 is used for intercomparison in this analysis. The MODIS instrument Justice et al. (1998) measures a radiance at the top of the atmosphere. These measurements must first be cloud-cleared and atmospherically corrected, following which, multiangle directional reflectances from both the Terra and Aqua MODIS sensors, over a period of 16 days, are accumulated for a location. From these directional reflectances, an appropriate RossThick LiSparse Reciprocal empirical kernel-based bi-directional reflectance distribution function (BRDF) model is estimated. The MODIS BRDF/albedo product is widely used and has been described by Lucht et al. (2000); Schaaf et al. (2002); and Stroeve et al. (2005). The retrieved BRDF is then integrated over all view zenith angle to calculate an intrinsic directional hemispherical reflectance (a black sky albedo) for the seven MODIS land bands. The BRDF model is further integrated over all possible illumination angles to produce a bihemispherical reflectance (or white sky albedo). The latest reprocessed operational Collection 6 MODIS daily BRDF/Albedo/NBAR products improve the temporal aggregation of snow observations (Wang et al., 2012, 2014) by using a daily measurement and triangulated filter to emphasis the nearest-day observations. The snow/non-snow status of day of interest is utilized for retrievals in Collection 6 instead of previous collection 5 strategy of only capturing snow measurements when snow cover represented the majority situation over the 16-day retrieval period. Of the seven available, bands 3, 4, and 1 are used here, see Table 1.





## 6 Discussion

In the following we evaluate differences between the UAS measured nadir reflectance and the MODIS NBAR product. Unless otherwise noted, the data refers to high quality flagged Collection 6 of the MODIS daily NBAR product (MCD43). We include also data from Collection 5 to demonstrate some of the marked improvements as well as evaluate the importance of quality
flagging.

The MODIS albedo product has been compared with in situ measurements in Greenland (e.g., Stroeve et al., 2005) and a number of other snow covered locations (Wang et al., 2014). The root mean square error between MODIS and in situ measurements was within ±0.04 (±0.07) for high quality (poor quality) flagged MODIS albedos. A high quality flag indicates that sufficient high quality surface reflectances to adequately sampled the full angular hemisphere were acquired, and a high
quality full inversion BRDF model was able to be developed and be used to produce NBAR and the instrinsic surface albedo quantities. For poor quality flagged retrievals a BRDF model could not be retrieved and a backup algorithm with a predetermined BRDF for that location had to be utilized. These intrinsic surface quantities are related to the surface structure and he albedos represent fully direct and fully diffuse values – therefore these need to be combined as a function of optical thickness to simulate the blue-sky albedos routinely captured with albedometers at surface tower locations (Lucht et al., 2000; Schaaf
et al., 2002; Román et al., 2010). In order to incorporate the full atmospheric effects, the full multiple scattering of Román et al. (2010) formation needs to be used over snow surfaces. While these analyses have been clearly valuable to the community, they do represent a complex intercomparison due to the nature of the measurement-model chain required to derive albedo. We reduce, by at least one degree the required modeling for our intercomparison by evaluating NBAR rather than albedo, which neither platform measures directly.

In general the agreement between the UAS nadir reflectance and MODIS NBARs are within the measurement uncertainties, Figs. 5 and 6. Systematically, the UAS measurements are slightly higher than the MODIS NBARs. For band 3 (blue) the MODIS NBAR is slightly smaller than the UAS reflectance: 0.967(0.965) versus 0.971 (0.978) for 5 Aug (6 Aug), see Table 2. The MODIS band 2 (green) NBAR is also slightly smaller, 0.966 (0.965) versus 0.974 (0.980) for 5 Aug (6 Aug). For the MODIS band 1 (red) NBAR the UAS reflectance is also slight smaller: 0.952 (0.950) versus 0.956 (0.967) for 5 Aug (6 Aug).
As expected from the refractive index of ice and NBAR calculations, Fig. 1, there is little wavelength dependence in the NBARs of bands 3 and 4, whereas there is an expected decrease in NBAR for band 1.

In Table 2 we also include a summary the MODIS Collection 5 data which is only retrieved once every 8 days and not filtered to weight the day of interest. The agreement between the UAS and MODIS version 6 is better than between the UAS and MODIS version 5 for bands 3 and 1. For band 4 there is better agreement between the UAS and MODIS version 5.
However, the differences are within the uncertainties, see below. Further, the standard deviation is generally smaller for the version 6 data.

Variability in the measurements accurately reflects the conditions at time of acquisition. Consistently, the standard deviations for all products are slightly larger for the flight on 5 Aug due to the more turbid atmospheric conditions. The variability is a product not only of the cloud cover, but also the wind speed, which potentially increased turbulence for the aircraft. We further





see strong support for the quality flagging of the MODIS products. The variations of the MODIS NBARs are larger for the pixels identified as low quality retrievals compared to the good quality retrievals (see standard deviations in parenthesis in Table 2).

The UAS measures the instantaneous up-welling radiance within 7° and the full hemisphere down-welling irradiance; the ratio of the two providing the reflectance. Whereas, the radiances measured by MODIS at the top of the atmosphere are atmospherically corrected to surface reflectance by means of radiative transfer modeling. These directional surface reflectances at a location – time weighted to the day of interest – are gathered over a 16 day period from both Terra and Aqua, and used to derive the BRDF for the full range of solar and viewing angles. The BRDF is then used to calculate an NBAR or an albedo for the seven MODIS bands. Here we have used the the BRDF to calculate a solar noon, nadir reflectance for comparison with the UAS. Hence, it is important to recognize the MODIS products are not instantaneous nadir reflectance measurements, but rather a calculation guided by data acquisition. Given this consideration, the correspondence is impressive. The standard deviation in the UAS data varies between 0.025 and 0.065. The standard deviation in high (low) quality MODIS data varies from 0.008 (0.009) to 0.015 (0.024). Given the smaller footprint, instabilities in the platform, and uncertainties related to the snow surface roughness, it is expected for the UAS derived reflectances to have higher standard deviation. Further, as these data are instantaneous measurements rather than an integrated model product, one should expect greater variance.

The footprint of the MODIS reflectance shown in this study is 500 m² at nadir, but has an effective footprint of 833 m x 613 m at the latitudes of this study (Campagnolo et al., 2016). The footprint of the UAS is circular with a diameter of about 30 m. Thus the UAS may be used to investigate MODIS sub-pixel variability. Where several UAS measurements are available within a pixel, there is considerably variability within a MODIS pixels, see data points with error bars in Figs. 5 and 6. However, this variability is within the uncertainty in the UAS data. Given more stable flight conditions, measurement of MODIS sub-pixel variability should be fully feasible with a UAS.

Optical satellite instruments require cloud free conditions to make NBAR estimates. This clearly limits the number of days available for NBAR measurements. The UAS is not limited the a cloud free sky, but may be used measure nadir reflectance also under cloudy conditions. However, as discussed below, the UAS must be below the cloud layer and not in it or a haze layer.

In Fig. 7 we evaluate the differences between Collection 5 and 6 MODIS products. The top two rows of panels are for 5 Aug, while the bottom two rows are for 6 Aug. The first and third row show Collection 5 compared with the UAS data, while the second and fourth rows show Collection 6. Two distinct features stand out. First, there is an improvement in the poorer quality magnitude estimates in Collection 6 as demonstrated by the systematic shift leftward of the data cluster from Collection 5 to 6. Most data fall below a reflectance of 1.0, whereas in Collection 5 several values were greater than 1. We note, a value of greater than 1 is not impossible, and in fact quite apparent for the UAS data, likely resulting from forward scattering driven by the sustruggi and expected at the scale of these measurements (30 m footprint). For the MODIS data, however, covering a $km^2$ footprint, one would expect reflectances to be more smoothed and values greater than 1 are expectedly rare. The second feature is a clear decrease in the variability of the data; both in terms of the overal spread, but also for the flagged values. There are fewer poor quality retrievals in Collection 6 resulting from the improved temporal retrieval frequency in the algorithm.





Regarding the quality and no data flagging we find the differences noted between Aug 5 and 6 clearly important, Fig. 8. The flight conditions were better (less clouds) on Aug 6 more MODIS data points have a good retrieval flag. Nevertheless the data points flagged as low quality have a spatial variation in agreement with the good quality points, compare yellow (low) and black (good) dots in Figs. 5 and 6 and the scatter plots shown in Fig. 7. Particularly for Collection 5, there is larger variation

in the low quality MODIS data, yet there is no support for this variation in UAS measurements. Thus overall, the MODIS algorithm appears to correctly discriminate good and low quality data.

We note significantly greater variability in the 5 Aug flight data. This is to be anticipated given the sky conditions, but the data provides a valuable reference for comparison with the MODIS data. Some features stand out from the flight, first being what appears to be a consistent decrease in reflectance from the start of the flight until 17:00 UTC, when the flight initiates

the SE leg. Overall the reflectance decreases in this period by almost 10%. The flight on the 6 Aug followed the same flight pattern, but no such drop in reflectance is present. There are several plausible explanations for the drop, including: a drop in surface reflectance; measurement error; presence of slightly absorbing particles in the atmosphere. As the drop was only seen on the 5 Aug and not on the subsequent day, and, further, the drop is not seen in the good quality MODIS data; therefore we rule out a change in the surface reflectance. The drop could be due to incorrect tilt correction. However we investigated this

thoroughly and find no feasible explanation that this error would be introduced on the 5 Aug flight but not on the 6 Aug flight. Additionally, as the drop occurred continuously through multiple legs and is then not seen at the end of the flight, we also rule out this explanation.

From Fig. 4 it is evident that some clouds were present during part of the flight on the 5 Aug. While these could have an impact on reflectance due to shadowing, a non-absorbing cloud will not change the surface reflectance as measured by the UAS.

On the other hand, if the UAS encountered an optically thin, slightly absorbing haze layer the UAS measured reflectance will drop. To quantify this drop radiative transfer calculations were made of the UAS reflectance with the UAS being at different flight altitudes. Cloudless and various haze conditons were considered. As shown in Fig. 9 the reflectance on a cloudless day (red line) does not depend on the altitude of the UAS. If an optically thin (optical depth 0.5) and slightly absorbing (single scattering albedo 0.95) haze layer of 1 km vertical thickness is included the UAS measured reflectance will be lower then the

surface reflectance (green line), and the difference will increase with increasing altitude.

The adopted haze layer optical property values are representative for those reported for Arctic haze, see for example Tsay et al. (1989); Hess et al. (1998) and Quinn et al. (2007). Increasing the absorption (single scattering albedo 0.9) increases the difference even more (blue line). The average flight altitude of UAS on the 5 Aug is indicated by the dotted line in Fig. 9. The drop seen in the UAS reflectance on the 5 Aug may thus be explained by the UAS entering an optically thin and slightly

absorbing haze layer. However, due to lack of additional measurements (aerosol properties) we can not prove this. But it is noted that during haze conditions reflectance measured by an airborne platform may be affected by the atmosphere between the platform and the surface. An ideal platform for surface reflectance observations would include aerosol observations as those presented in Bates et al. (2013).

Covering the same flight path on the 6 Aug, the UAS data and MODIS products agree remarkably well. As mentioned

earlier, we note greater variability in the UAS data. And in the case of these observations, this variability is likely resulting





from several factors. For one, the UAS is measuring a smaller area, and thus there is far less smoothing of the data. Certainly surface roughness and forward scattering play a role, but despite best efforts to select only stable periods of flight, there is likely error introduced to the data from platform stability. Nonetheless, the consistent feature is that the UAS observations are larger overall than the MODIS products. Given the nature and temporal smoothing of the MODIS products, we believe this is a real artifact and that the MODIS data may in fact provide values slightly lower than actual reflectances that would be observed instantaneously. We also note that the UAS platform has immense potential to provide greater insight into the sub-pixel variability of the MODIS products – which is likely significant.

## 7   Conclusions

During July-August, 2010, several UAS flights were made over Summit, Greenland. Two of the flights were designed to cover the MODIS swath and were made close in time to the MODIS overpass. The UAS measured the up- and down-welling radiation between 320-950 nm with a resolution of 3.3 nm. In this analysis we have made a direct comparison of reflectance as measured by the UAS with the MODIS NBAR product. The main findings are:

– The UAS and MODIS reflectances for band 3, 4 and 1 agree within their uncertainties. However, due to the larger footprint and temporal smoothing of MODIS, the product provides a slightly lower overall reflectance.

– Sub-pixel variability of MODIS reflectance products is potentially significant. Further work should be conducted to evaluate the magnitude and subsequent impacts to modeling of greater variability at the sub-kilometer scale.

– Consistent with theory, the UAS and MODIS reflectance measurement show a decrease between band 3 and 4 and band 1. This wavelength dependence agrees with that expected from the refractive index of ice.

– Even at the low elevations flown typically by UAS, reflectance measurements may be influenced by haze if present at and/or below the flight altitude of the UAS.

– The UAS platform is proven as a capable resource to collect reflectance measurements over an extensive region and provides a reliable resource for evaluating spatial variability of reflectance for Summit, Greenland and the surrounding area.

Of significance in this evaluation, and any intercomparison, is the concept of 'Truth'. Neither of the platforms presented provide a perfect measure of NBAR, but this is a much more direct intercomparison than would be with albedo which would require several further assumptions. MODIS and the UAS platforms attempt to provide an accurate characterization of NBAR and nadir reflectance, respectively. However, MODIS, while collecting direct radiance measurements, requires a fairly complex model and data assimilation chain to provide a product – ultimately produced from a BRDF model. The UAS is challenged due to platform instabilities and expected instrumental errors. The nature of reflectance observations is further challenged by the fact that it is not a simple property of a surface (e.g. snow), but rather a product of a system. The system includes many temporally



varying factors such as solar zenith angle and azimuth, but also more complex processes including atmospheric scattering, surface roughness, and snow conditions. The time scales of variability of these processes differ and are complex to represent consistently. Given the importance of this parameter and the general derivation of albedo from these observations, it is critical that we understand well the expected variability. The UAS platform provides a unique opportunity to collect observations

that are more representative spatially for applications where point-based measurements are used; for instance, the validation of satellite-based remotely sensed measurements and 'grid'-based climate and earth system models. This work demonstrates the feasibility of collecting these observations, but also exemplifies the challenges associated with benchmarking different observations. Furthermore, the observations presented herein were collected over a range offering relatively low variability in reflectance. To further increase our current understanding of the reflectance of the cryosphere and its development, more UAS

measurement campaigns at other locations and surface conditions are warranted.

## 8   Data availability

MODIS data are available from http://modis.gsfc.nasa.gov/. Surface irradiance data for Summit Greenland are available from https://nsidc.org/. The UAS data are available from the authors upon request.

*Acknowledgements.* This work was conducted within the Norwegian Research Council's NORKLIMA program under the Variability of

Albedo Using Unmanned Aerial Vehicles (VAUUAV) project (NFR no. 184724), Hydrologic sensitivity to Cryosphere-Aerosol interaction in Mountain Processes (HyCAMP) project (NFR no. 222195), and with additional support from NFR no. 196204 (RISCC) and NFR no. 195143 (Arctic-EO). We acknowledge the Greenlandic Home Rule Government for permission to work in Greenland; the $109^{th}$ NY ANG for unsurpassed air transport; CH2MHill Polar Resources and the on-site science technicians for superb logistical support; the GEOSummit Science Coordination Office (SCO, PLR-1042531) for providing contacts and access to data; and overall the support from the Division of

Polar Programs at the National Science Foundation. Konrad Steffen provided the shortwave radiation data from Summit while NOAA-ESRL (Brian Vasel and Tom Mefford) provided wind speed and direction.



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





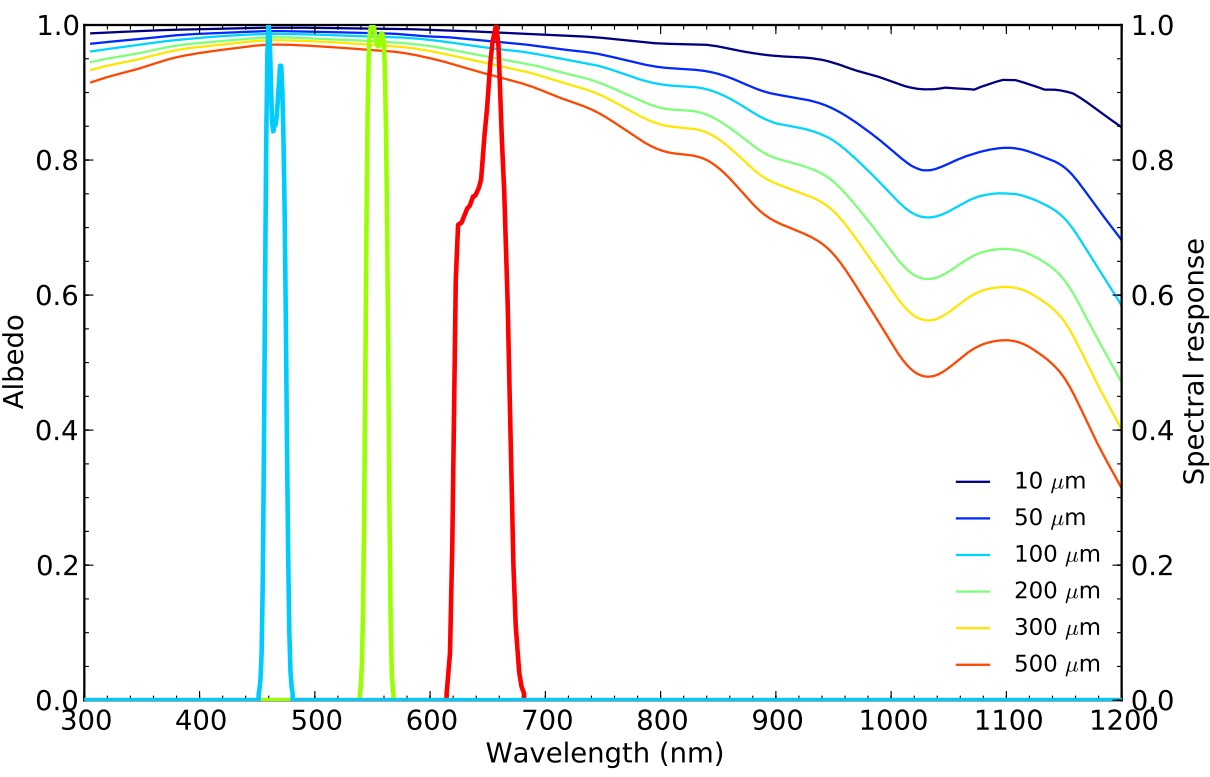

**Figure 1.** The pure snow albedo as a function of wavelength for snow grain sizes between 10-500 $\mu$m. Also shown in the thicker lines are the MODIS spectral response for band 3 (blue), 4 (green) and 1 (red). The spectral response data were obtained from http://mcst.gsfc.nasa. gov/calibration/parameters.





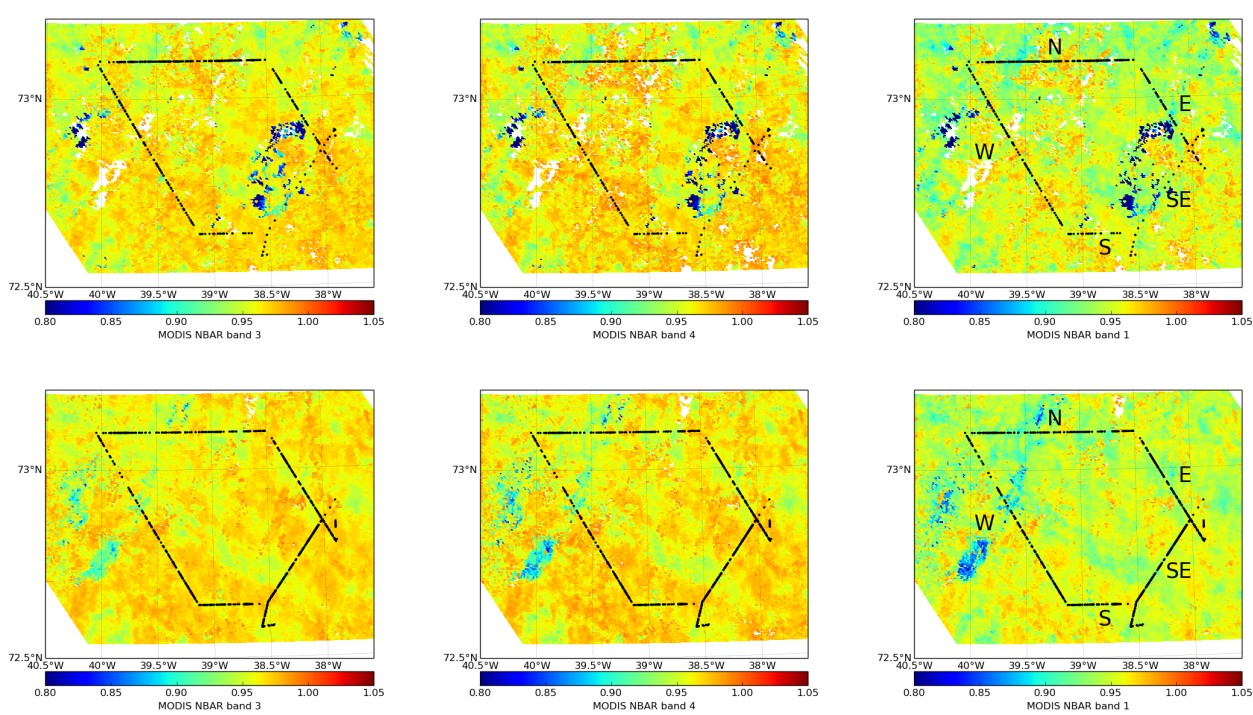

**Figure 2.** MODIS blue (band 3), green (band 4) and red (band1) NBARs for Aug 5 (upper row) and Aug 6 (lower row), 2010. The black dots represent UAS data which were recorded during stable flight conditions. White areas indicate missing data.





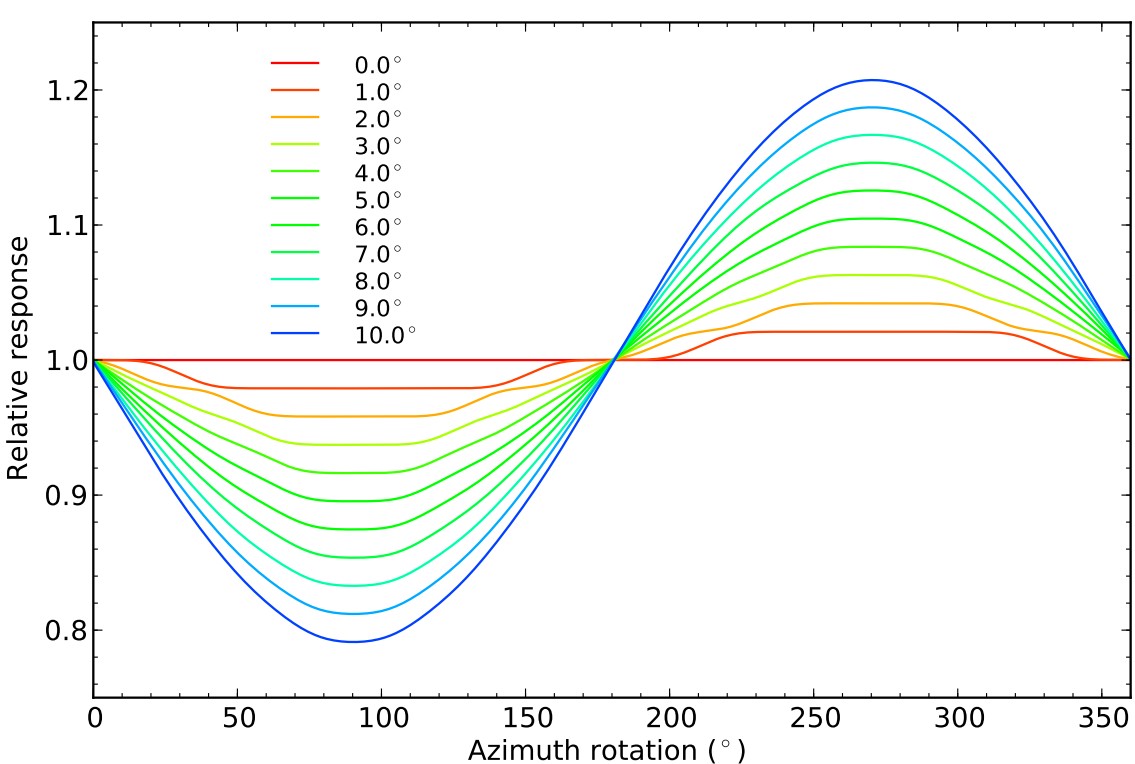

**Figure 3.** The relative response when tilting (various coloured curves) and rotating around azimuth an irradiance sensor on-board a platform 300 m above a snow surface. The solar zenith (azimuth) angle is 55.66° (0°). The wavelength is 465 nm.

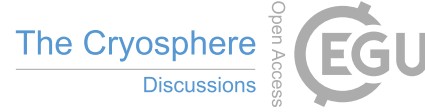



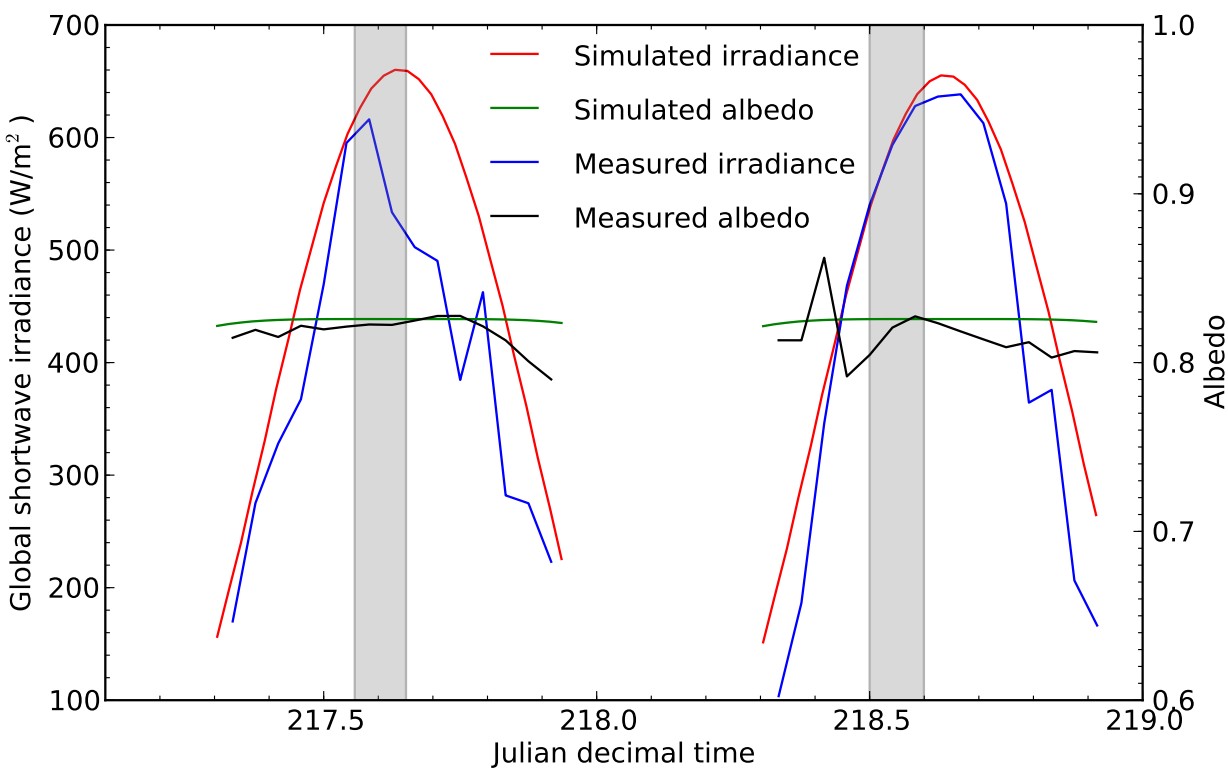

**Figure 4.** The measured (blue) and the simulated cloudless sky (red) downwelling short wave radiation for Summit, Greenland on Aug 5 (day 217) and 6 (day 218), 2010. The measured (black) and simulated (green) shortwave broadband albedos are also shown. The grey shaded areas indicate the time when the UAS was flying.





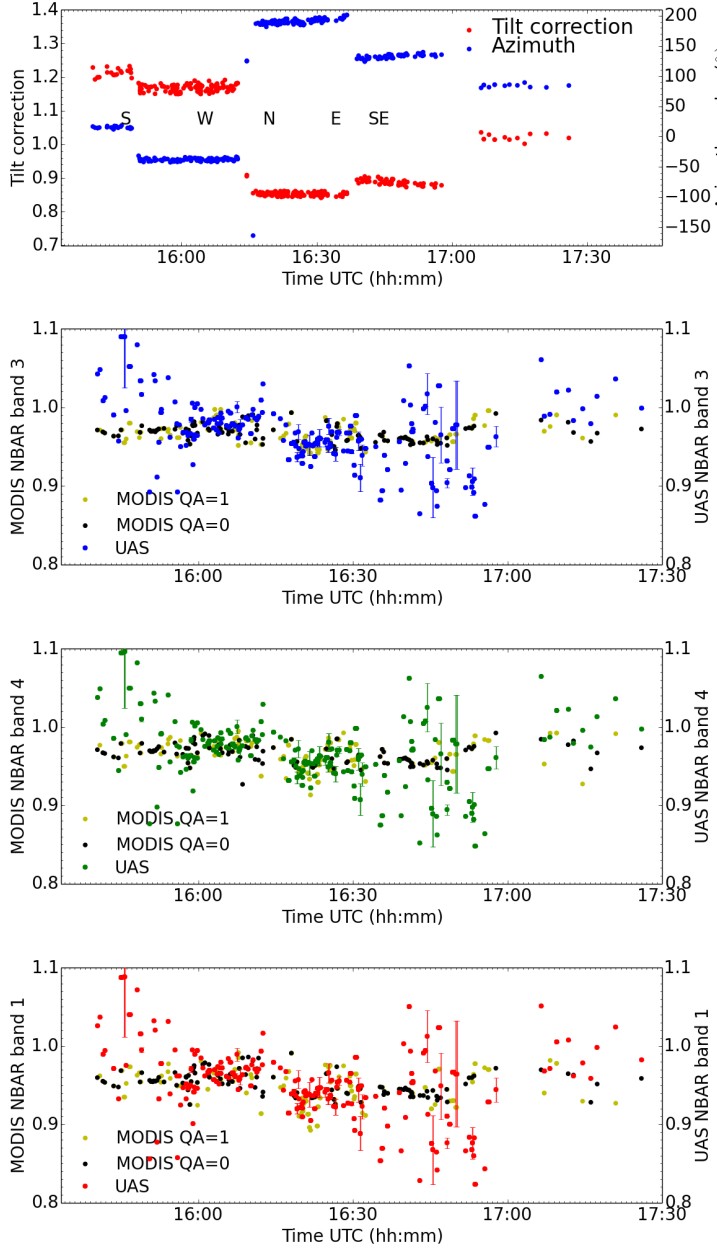

**Figure 5.** Top panel: The azimuth angle of the UAS (blue dots) and the tilt correction factor $R^l/R^t$, Eq. 3 (red dots). The various flight track elements are identified by letters and corresponds to the flight tracks in Fig. 2. Second-fourth panels: The MODIS (black=good quality and yellow=low quality dots) and UAS band 3 (blue),4 (green) and 1 (red) NBARs as a function of time. In the case of several UAS data points within one MODIS pixel, the UAS data have been grouped together and presented as a dot with standard deviation. All data from the flight on Aug 5, 2010.





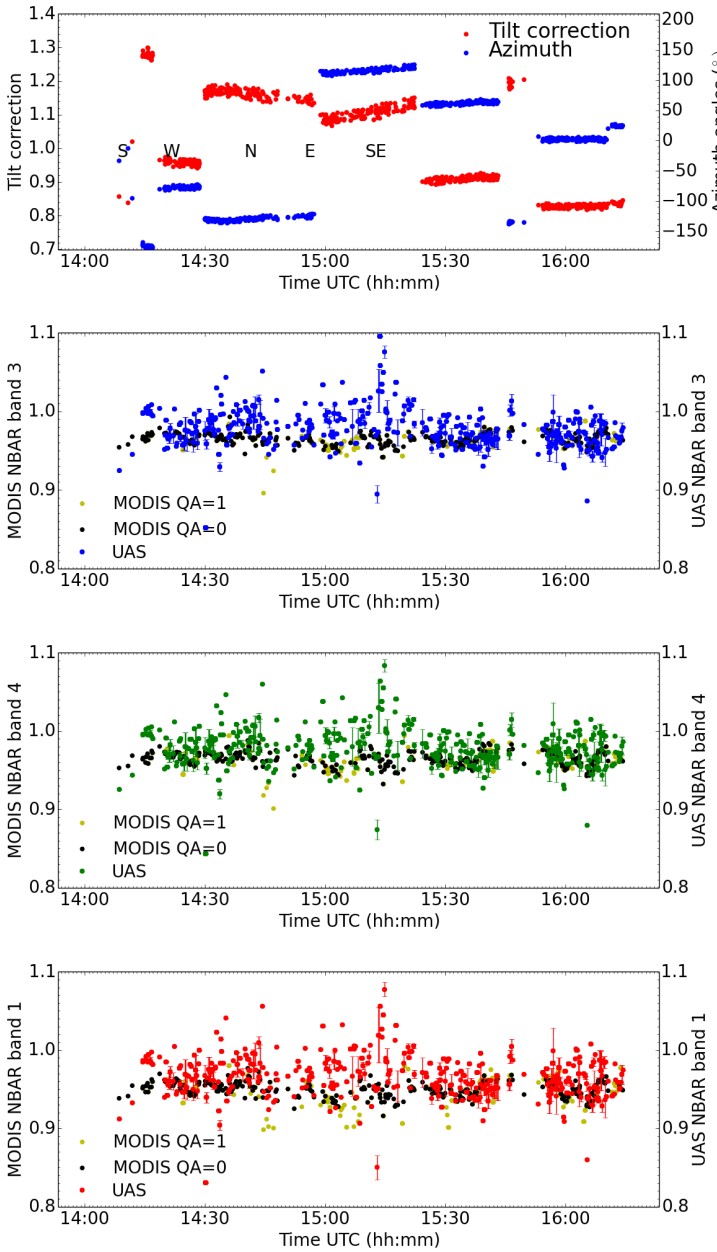

**Figure 6.** Top panel: The azimuth angle of the UAS (blue dots) and the tilt correction factor $R^l/R^t$, Eq. 3 (red dots). The various flight track elements are identified by letters and corresponds to the flight tracks in Fig. 2. Second-fourth panels: The MODIS (black=good quality and yellow=low quality dots) and UAS band 3 (blue),4 (green) and 1 (red) NBARs as a function of time. In the case of several UAS data points within one MODIS pixel, the UAS data have been grouped together and presented as a dot with standard deviation. All data from the flight on Aug 6, 2010.



**Figure 7.** The MODIS (black=high quality and grey=low quality) versus UAS NBAR for bands 3, 4 and 1. Left column is data for band 3, middle column is band 4 data and right column is data for band 1. Rows 1 and 3 are MODIS version 5 and rows 2 and 4 MODIS version 6. Rows 1-2 (3-4) are data from the flight on Aug 5 (6).



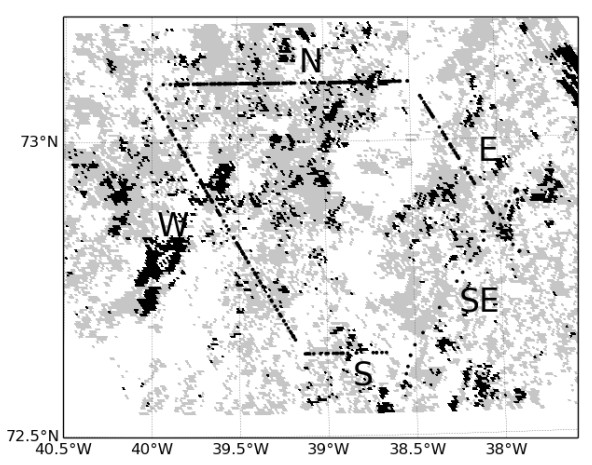 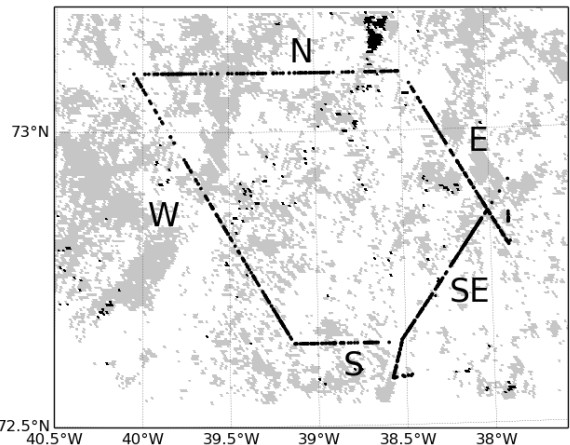

**Figure 8.** MODIS quality flags for Aug 5 (left panel) and Aug 6 (right panel), 2010. Grey color indicates low quality retrieval, black indicates no retrieval. White areas have high quality retrieval flags.




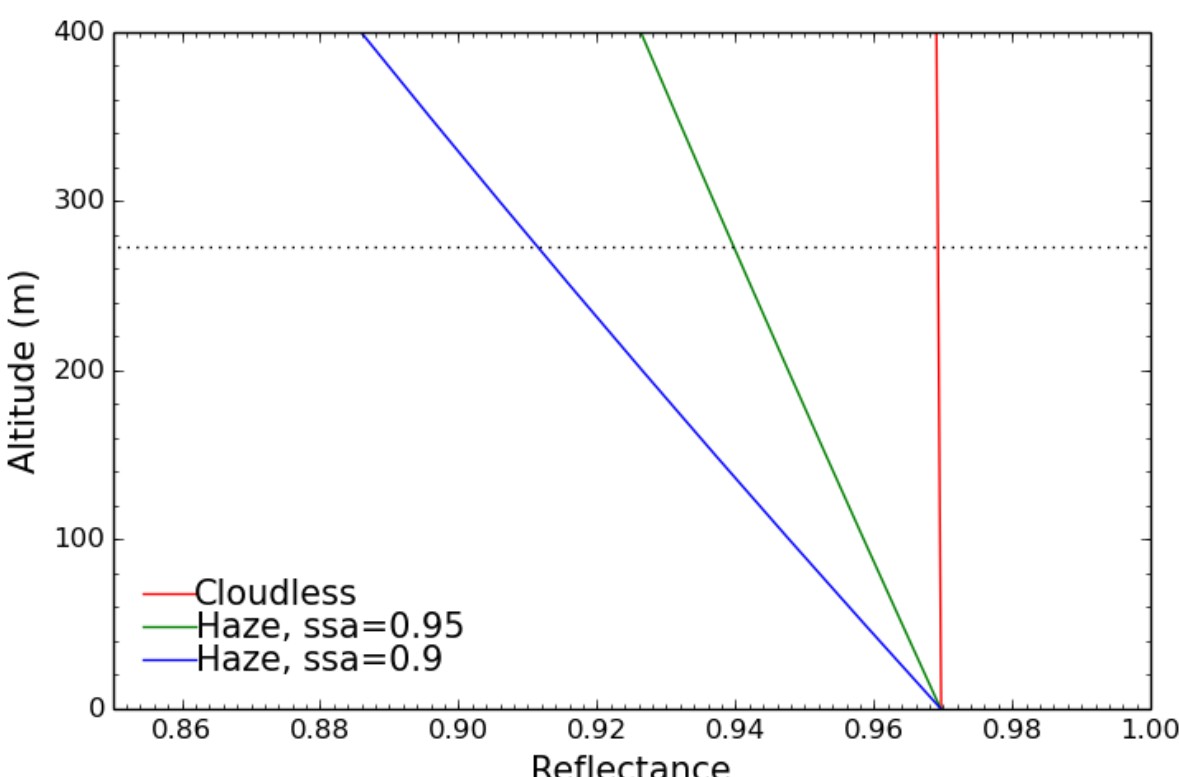

**Figure 9.** Simulated UAS reflectance for MODIS band 1 as a function of flight altitude for various sky conditions. See text for details. The dotted horizontal line indicates the average flight altitude of the UAS on the 5 Aug. The solar zenith angle is 58° corresponding to the values during the flight on the 5 Aug, see Table 2.





**Table 1.** The minimum ($\lambda_{min}$) and maximum ($\lambda_{max}$) wavelengths covered by MODIS bands 1, 3, and 4. Bandwidth for MODIS obtained from http://modis.gsfc.nasa.gov/about/specifications.php.

| Band | $\lambda_{min}$ (nm) | $\lambda_{max}$ (nm) |
|------|-----------|-----------|
| 3 | 459 | 479 |
| 4 | 545 | 565 |
| 1 | 620 | 670 |



**Table 2.** Information for the two UAS MODIS route overpass flights. For the NBAR values the average value is given together with the standard deviation in parenthesis.

|  | Aug 5 | | Aug 6 | |
| --- | --- | --- | --- | --- |
| Start of flight (hh:mm:ss, UTC) | 15:22:55 | | 14:00:28 | |
| End of flight (hh:mm:ss, UTC) | 17:37:27 | | 16:23:34 | |
| Mean pitch (std) angle (°) | 6.05 (0.55) | | 7.32 (0.52) | |
| Mean roll (std) angle (°) | -4.43 (0.48) | | 5.40 (0.54) | |
| Weather | Overcast | | Mostly cloudless | |
| Solar zenith angle (°) | 60-56 | | 55–58 | |
| Solar azimuth angle (°) | -46– -17 | | 11– -27 | |
| Wind speed (m/s) | 6-7 | | 1.5-3.5 | |
| Wind direction (°) | ≈135 | | 140-170 | |
| # UAS spectra | 497 | | 882 | |
| MODIS quality flag | 0 (Good) | 1 (Low) | 0 (Good) | 1 (Low) |
| MODIS ver 5, NBAR band 3 (blue) | 1.005 (0.012) | 1.005 (0.015) | 1.017 (0.011) | 1.017 (0.012) |
| MODIS ver 6, NBAR band 3 (blue) | 0.967 (0.010) | 0.965 (0.014) | 0.965 (0.008) | 0.966 (0.009) |
| UAS band 3 | 0.971 (0.056) | | 0.978 (0.025) | |
| MODIS ver 5, NBAR band 4 (green) | 0.978 (0.011) | 0.967 (0.024) | 0.988 (0.010) | 0.982 (0.019) |
| MODIS ver 6, NBAR band 4 (green) | 0.966 (0.013) | 0.964 (0.016) | 0.965 (0.009) | 0.965 (0.012) |
| UAS band 4 | 0.974 (0.064) | | 0.980 (0.026) | |
| MODIS ver 5, NBAR band 1 (red) | 0.939 (0.0123) | 0.956 (0.027) | 0.948 (0.011) | 0.937 (0.022) |
| MODIS ver 6, NBAR band 1 (red) | 0.952 (0.0146) | 0.947 (0.020) | 0.950 (0.010) | 0.949 (0.014) |
| UAS band 1 | 0.956 (0.065) | | 0.967 (0.028) | |