# Peer review of "Unmanned Aerial System nadir reflectance and MODIS Nadir BRDF-Adjusted surface Reflectances intercompared over Greenland"

_The Cryosphere, 2016_

## Referee Comment (RC1) · Anonymous Referee #1 · 13 Feb 2017

**Review of 'Unmanned Aerial System nadir reflectance and MODIS Nadir BRDF-Adjusted surface Reflectances intercompared over Greenland' by Burkhart et al, submitted to The Cryosphere.**

The authors present observations of narrowband reflectance corresponding to blue, green and red wavelengths made by UAS on two consecutive days at Summit, Greenland. They compare these observations to the MCD43 product produced from spatio-temporally coincident MODIS overpasses. The paper essentially outlines proof-of-concept in acquisition of narrow-band reflectance from UAS and comparison to a satellite-borne sensor. Overall they find that (a) there is very good agreement between the reflectance observations captured by the two techniques, but with large sub-MODIS pixel variability; and (b) that while UAS platforms are very promising in terms of their ability to acquire data, they require a significant amount of post-processing and quality-filtering to deliver reliable results.

In this review I write as an end-user of several MODIS reflectance and albedo products and with some secondary involvement in the use of UAS for understanding sub-pixel variability, but with the caveat that I am not an expert in the radiative transfer physics or remote sensing optics.

To my knowledge this is the first direct comparison of MODIS vs UAS narrowband reflectances, as opposed to attempts to convert UAS-acquired data to satellite-derived broadband albedo (e.g. Ryan et al., 2016). The paper is therefore an original and novel contribution in what is still a relatively immature research area. As such, a key strength of the paper is the relatively exhaustive level of methodological detail. Coverage of post-acquisition processing and quality control is seems excellent, especially the range of modelling experiments carried out to establish UAS sensitivities. I welcomed the comparison to both collections 5 and 6 of the MCD43 product – the results here provide a clear independent estimate of how much the re-processing has improved in collection 6 data. I cannot think of any other sensitivities or methodological issues which the authors should have accounted for.

I see no substantive omissions, mistakes or assumptions in this manuscript. However, there are several problems with the presentation and the fluency of the language which hinder readability of the manuscript. I got the sense that some methodological details were quite repetitive through the manuscript so it would be worth checking if clarity could be improved here.

Detailed comments:

P3, L8-34: structure confused in places, e.g. on L12-13 you introduce the 210 km track at the end of a paragraph about the novelty of the technique you are using. Move to the following paragraph to given a 1-para summary of your methods. Last para on this page – very wordy, do you need link to the overall project you are part of?

General observation: 'complex' appears often, this isn't a very precise word and so should be removed where possible. P2, L19-25. You mention ground-based measurements – what is being compared to what?

P8, Radiative transfer simulations: I'm unclear what these simulations are actually used for?

P10, L12-19 – discussion about blue-sky albedos needs to be rephrased for clarity, I'm not exactly sure what point you are making here.

P11, L4-11 – 'given this consideration, the correspondence is impressive' – not entirely sure how the consideration (presumably about MODIS but this isn't clear) maps onto the UAS data?

P11, L25-27 – this is essentially the figure caption, remove.

There are too many examples of bad grammar and typesetting for me to list here. I recommend proof-reading by a fluent English speaker. In terms of typesetting, the most glaring problem is that often the references appear as "statement about x Burkhart et al (2016)" when they should appear as "statement about x (Burkhart et al, 2016).

The figures are generally of good quality.

Fig 1: move away from rainbow colour palette for the different snow grain sizes, this is especially confusing on a plot with wavelength as the x axis. Suggest move to monotone colour palette.

Fig 2: change colourmap of images to something meaningful, i.e. a monotone linear colour ramp. In addition rainbow colourmaps present colourblind readers with significant difficulties and for this reason alone should not be used.

Figs 5 and 6: the transect direction labels are rather unclear. Can you add section dividers or equivalent to segment the different portions of the flight?

Figs 5 and 6: There is insufficient difference in colour between the UAV measurements for band 3 versus MODIS QA=0. Please change.

Figs 5 and 6: would suggest labelling MODIS QA as 'good' and 'bad', so the reader doesn't have to remember what 0 and 1 are – to me they are arguably the 'wrong' way around!

Fig 7: the labelling here needs improvement. Label each row with exactly what it is showing (i.e. date, MODIS collection) rather than leaving it to the reader to work out from the caption.

Table 1: the contents of this is essentially shown in Figure 1 and I therefore suggest that this could be dropped to save space.

---

## Referee Comment (RC2) · Anonymous Referee #2 · 6 Mar 2017

This is a novel, pioneering and worthwhile comparison of nadir surface reflectance from a Unmanned Aerial System in comparison with satellite (MODIS) albedo data. This study should be of wide interest to the Greenland Ice Sheet glaciological and remote sensing communities. In general, I concur with the comments of Referee 1. In addition, the writing style can occasionally be tightened/ page 1, line 7: rephrase "allowing to integrate directly to". p.1, l.7: "The data PROVIDE a unique opportunity to...". p.2, l.12 should be "INTERGOVERNMENTAL Panel on Climate Change". p.3, l.6: "Ryan et al. (2016) HAVE attempted to..." p.3, l.25: "we discovered THAT the application of UAS..." p.11, l.29: "We note a value of..." (no comma). p.13, ll.4 & 5: "the UAS observations are larger overall" AND "the MODIS data may in fact provide values

slightly lower than..." - give actual values/differences and say whether these are significant. p.14, l.2: "time scales" -> "timescales". In addition, I recommend a tabulation of some of the more quantitative aspects of the comparison of nadir reflectance from the two types of platform, that are reported on pages 9 & 10 and in Figures 5 & 6.

———————————————

---

## Author Comment (AC1) · 1 Apr 2017

**1 General Response**

The reviewer provided helpful comments regarding the context and placement of the manuscript, and helped to highlight where some points were felt to be redundant. We appreciate the feedback and have taken it into consideration throughout our revision process.

[Figure]

**2 Detailed comments:**

*P3, L8-34: structure confused in places, e.g. on L12-13 you introduce the 210 km track at the end of a paragraph about the novelty of the technique you are using. Move to the following paragraph to given a 1-para summary of your methods. Last para on this page – very wordy, do you need link to the overall project you are part of?*
We have tried to rewrite and clean the wording here. We feel mentioning the project is quite relevant as it helps place our work in the context of not only albedo variability, but also the level of accuracy we sought as we attempted to make measurements of relevance to the 'aerosol deposition' story.

*General observation: 'complex' appears often, this isn't a very precise word and so should be removed where possible.*
Thank you, we've gone through now and adjusted our working accordingly.

*P2, L19-25. You mention ground-based measurements – what is being compared to what?*
In general, satellite sensor data to ground-based measurements. Further in the text, we describe for the different studies referenced what is being compared. To some degree, this is the point we attempt to make, that many 'albedo' validation studies are comparing the measured blue sky albedos with satellite radiances (and their associated model chains).

*P8, Radiative transfer simulations: I'm unclear what these simulations are actually used for?*
We hope this is further clarified now in the text. These are used primarily to address sensitivity to the haze layer (Fig.9) as well as our evaluation of the platform sensitivity to orientation (Fig. 3).

*P10, L12-19 – discussion about blue-sky albedos needs to be rephrased for clarity, I'm not exactly sure what point you are making here.*
This is now clarified in the text.

*P11, L4-11 – 'given this consideration, the correspondence is impressive' – not entirely sure how the consideration (presumably about MODIS but this isn't clear) maps onto the UAS data?*
We have adjusted the sentence to clarify the 'correspondence'.

*P11, L25-27 – this is essentially the figure caption, remove.*
Good point. Done.

*There are too many examples of bad grammar and typesetting for me to list here. I recommend proof-reading by a fluent English speaker. In terms of typesetting, the most glaring problem is that often the references appear as "statement about x Burkhart et al (2016)" when they should appear as "statement about x (Burkhart et al, 2016).*
Thank you. Hard to swallow, but perhaps I've lived too long within a foreign language environment, that I am now losing my own! Regardless, you are correct. I found these errors now and believe we have addressed them all.

**3   Figure Commments:**

*The figures are generally of good quality.*
Thank you.

*Fig 1: move away from rainbow colour palette for the different snow grain sizes, this is especially confusing on a plot with wavelength as the x axis. Suggest move to monotone colour palette.*
We've adjusted this now. We recognize the blue,green,red scheme is not optimal for color blind individuals, but the colors reflect the colors of the MODIS bands and are not essential in this context to distinguish from one another, but rather show the general shape of the response functions.

[Figure]

*Fig 2: change colourmap of images to something meaningful, i.e. a monotone linear colour ramp. In addition rainbow colourmaps present colourblind readers with significant difficulties and for this reason alone should not be used.*
We've addressed this now.

*Figs 5 and 6: the transect direction labels are rather unclear. Can you add section dividers or equivalent to segment the different portions of the flight?*
This should be more clear now. We've changed Figure 8 and highlighted the direction of the flight as well as adding some further descriptive information in the caption.

*Figs 5 and 6: There is insufficient difference in colour between the UAV measurements for band 3 versus MODIS QA=0. Please change.*
Done.

*Figs 5 and 6: would suggest labelling MODIS QA as 'good' and 'bad', so the reader doesn't have to remember what 0 and 1 are – to me they are arguably the 'wrong' way around!*
We only follow protocols as defined by NASA for their products. We've tried to make this more clear.

*Fig 7: the labelling here needs improvement. Label each row with exactly what it is showing (i.e. date, MODIS collection) rather than leaving it to the reader to work out from the caption.*
This is corrected.

*Table 1: the contents of this is essentially shown in Figure 1 and I therefore suggest that this could be dropped to save space.*
We have removed Table 1 and placed the reference to the MODIS specifications in the text.

---

## Author Comment (AC2) · 1 Apr 2017

**1    General Response**

We appreciate the reviewers acknowledgement of the contribution this work is making toward MODIS validation and improvements as well as the development of UAS for scientific applications.

[Figure]

**2 Specific Comments:**

*In addition, the writing style can occasionally be tightened/ page 1, line 7: rephrase "allowing to integrate directly to".*
Corrected.

*p.1, l.7: "The data PROVIDE a unique opportunity to...".*
Corrected.

*p.2, l.12 should be "INTERGOVERNMENTAL Panel on Climate Change".*
Corrected.

*p.3, l.6: "Ryan et al. (2016) HAVE attempted to..."*
Corrected.

*p.3, l.25: "we discovered THAT the application of UAS..."*
Corrected.

*p.11, l.29: "We note a value of..." (no comma).*
Altered text.

*p.13, ll.4 & 5: "the UAS observations are larger overall" AND "the MODIS data may in fact provide values slightly lower than..." – give actual values / differences and say whether these are significant.* Addressed.

*p.14, l.2: "time scales" -> "timescales".*
Corrected.

*In addition, I recommend a tabulation of some of the more quantitative aspects of the comparison of nadir reflectance from the two types of platform, that are reported on pages 9 & 10 and in Figures 5 & 6.*
Per RC1 we have replaced Table 1 with a new table tabulating the information discussed in the text.

---

## Author Comment (AC3) · 1 Apr 2017

As a supplement here we include the diff latex document. However, please note that table 1 caused problems when building the diff document. Therefore it is provided as a figure here. Note that subsequently in the diff document, Table 2 is referred to as Table 1.

Please also note the supplement to this comment:
http://www.the-cryosphere-discuss.net/tc-2016-264/tc-2016-264-AC3-supplement.pdf

[revised manuscript text omitted]

---

## Author Response (AR2)

**UiO **Department of Geosciences**
**University of Oslo**

Prof. Edward Hanna
Editor
The Cryosphere
Copernicus Office

Date:      17 May, 2017
Your ref:   tc-2016-264

**Dear Professor Hanna**
Thank you for the comments regarding our submission. I believe we have now addressed the revisions you have requested, including:

- Changes to wording regarding "larger" and "slight"
- clarification of the atmospheric processes which we investigate
- addition of Table references

We look forward to your response and appreciate the effort and work of the Copernicus editorial staff.

Kind Regards,

John F. Burkhart
Associate Professor

 on behalf of the VAUUAV science team.
encl: Abstract
encl: Final Manuscript

[Figure]

Department of Geosciences | Physical Geography and Hydrology | Phone:   (47) 96 82 50 11
| Sem Saelands vei 1 | Fax:     (228) 524 10
| 0371 Oslo | E-mail:
| | john.burkhart@geo.uio.no
| | Www:    www.geo.uio.no